# The Biological Processes of Chloride Ions Removal from the Environment

**Elżbieta Sobiecka** 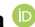

Institute of Natural Products and Cosmetics, Faculty of Biotechnology and Food Sciences,
Lodz University of Technology, ul. Stefanowskiego 2/22, 90-537 Lodz, Poland; elzbieta.sobiecka@p.lodz.pl

**Abstract:** Chlorine is one of the macronutrients commonly found in nature. The natural cycle of this element can be destabilized by human activities and causes negative effects in the environment. To come back into a natural balance, various biological processes of water and soil remediation have been investigated. The purpose of the presented research focused on two chlorine conversion processes: (1) anaerobic dechlorination of polychlorinated biphenyls (PCBs) found in transformer oil provided by consortia of microorganisms originating from a wastewater sedimentation tank and (2) chloride elimination from aquatic environments by commercial mixtures of microorganisms in aerobic conditions. Dechlorination allowed the conversion PCB molecules to less-toxic compounds and significantly influenced contamination in the range of 15 to 76%. In the second process, the decrease in chloride ions did not exceed 14%. Both the consortia of microorganisms and biological commercial mixtures used in this study were able to decrease the chloride ion concentration in the investigated aquatic solution.

**Keywords:** chloride ions; biodegradation process; dechlorination; consortia and commercial mixtures of microorganisms

## 1. Introduction

Chloride anions are mainly found in oceans, seawater, in the waters of salt lakes, rivers, and surface waters. They are also found in the extensive underground deposits formed by NaCl that have appeared because of the drying up of seas. It occurs as the forms of minerals such as halite (rock salt), sylwin (potassium chloride), and carnallite (hydrated double potassium chloride) [1].

Chlorides are present in all living organisms where they are needed to maintain an appropriate osmotic pressure. In the human body, it is the main component of the physiological solution. Chlorides are a basic ingredient in animal and plant nutrition [2].

Chlorine is one of the macronutrients commonly found in nature. The natural cycle of this element helps to maintain a balance in nature which can be destabilized by human activities. Chlorine compound accumulation in a small area causes negative effects in the environment which requires remediation to achieve a natural balance again.

Water contamination by chlorine compounds can be caused by inflow from landfills [3]. Calcium and sodium chlorides are used to defrost roads, which causes a large increase in chlorides in the environment [4].

Over the years, agglomeration practices have influenced water irrigation changes or leachate recirculation changes in urban and agricultural hydrological cycles. In fact, new challenges have appeared in urban and landfill management. Advanced technologies and modeling approaches allow the creation of hydrological modelling tools to mitigate the contamination of groundwater and deep aquifers by various ions such as chlorides [5]. Organic and inorganic pollutants can directly influence the changes in natural element balance e.g., carbon, oxygen, nitrogen, or phosphorous. Dissolved organic carbon is one of the most important elements for microorganisms in their metabolic pathways. In

contaminated sites, the balance between macro- and micro-elements can be destroyed if the concentrations of pollutants increase. In addition, the changes in the amount of dissolved organic carbon from surface and groundflow water may significantly influence biogeochemical cycles [6].

Agricultural production also is an important source of water pollution through releasing chloride ions originating from fertilizers [7]. The rock salt used in farms and on pastures as animal licks also causes an increase in chlorine in ecosystems [8,9].

Chlorine compounds are used in various branches of industries. One example is the power industry, where transformers filled with oils containing commercial mixtures of polychlorinated biphenyls (PCBs) have been used for years. Defective operation of industrial equipment as well as various types of accidents may lead to soil and water contamination. In this way, chlorine-containing compounds can pollute the environment [10].

Although this element is very widespread around the globe and necessary for the functioning of organisms, its high concentration in aquatic and terrestrial ecosystems may pose a threat to the proper functioning of fauna and flora. Therefore, researchers for many years have been conducting experiments aimed at the effective removal of chlorine from places where its accumulation exceeds permissible doses.

There are biological, chemical, and physical methods that are used as fundamentals to study the environmental remediation of polluted sites [11–13]. Biological processes such as biodegradation and bioremediation are environmentally friendly but often require more time than others to move pollutants from the solution. Biodegradation process technologies use autochthonous microbial communities capable of metabolizing contaminants [14,15]. Microorganisms (bacteria or fungi) which naturally appear in polluted sites are screened and studied in order to biodegrade toxic compounds and their metabolites. There are many species of bacteria which provide the biodegradation processes in polluted sites. The processes that appear during various stages of the chlorine cycle in nature require different oxygen conditions. The biological processes of chlorine pollutant degradation on water surfaces or on soil surfaces can be provided by aerobes. Others remove chlorine from molecules in the anaerobic process of dechlorination. Some microorganisms use various ions to provide redox reactions in a deeper level of the soil where oxygen is limited. Two of the common microorganisms that participate in dechlorination processes are sulphate-reducing bacteria (SRB) and denitrifying bacteria (DB) [16,17]. The most effective species are used to prepare a consortium of different microorganisms which might result in the establishment of new commercial mixtures to remove pollutants from the environment in catabolic reactions [18].

The chemical methods used in water cleaning processes from toxic compounds focus on the investigation of adsorption techniques which offer good results for heavy-metal-ion removal. Many adsorbent hydrogels such as N,Ndimethylacrylamide (DMAA)-based hydrogels have been reported as effective tools in such processes [19,20]. Physical methods also can be used in soil and water remediation, but the high cost of the apparatus used in these methods has meant that these methods are not popular.

The aim of this study was to present the effectiveness of chlorine compound conversion in two processes that can appear during the natural chlorine cycle. The first one was a dechlorination process of organic pollutants containing chlorides in their molecules. The compounds used in this study were polychlorinated biphenyls found in transformer oil. The process was provided by consortia of microorganisms originating from a petrochemical wastewater sedimentation tank. The second process tested the ability of a commercial mixture of microorganisms to move chloride ions from an aquatic environment.

## 2. Materials and Methods

### 2.1. Biological Materials

2.1.1. Dechlorination Process

The sulphate-reducing bacteria (SRB) and denitrifying bacteria (DB) present in the primary sediments originating from the petrochemical wastewater sedimentation tank

(Door's tank) of the Petrochemistry Plant in Plock, Poland, were used in the dechlorination process.

The activation of the microorganism consortia was carried out in mineral medium: $(NH_4)_2SO_4$ 1.0 g/dm$^3$, $Na_2HPO_4$ 0.2 g/dm$^3$, $KH_2PO_4$ 0.2 g/dm$^3$, $MgSO_4$ 0.02 g/dm$^3$, $FeSO_4 \cdot 7\,H_2O$ 0.01 g/dm$^3$, and $Na_2CO_3$ 0.02 g/dm$^3$ at a temperature of 30 °C. After one day, 10 mL/dm$^3$ of inoculum was added to cultures containing polychlorinated biphenyls (PCBs).

### 2.1.2. Chloride-Removing Process

A commercial mixture of microorganisms, BIOACTIV PH, was purchased from the EKOB-TBA Sp. z o.o. Company, Gliwice, Poland. It was used to eliminate the chlorine ions from two aquatic media.

The activation of the commercial mixture was carried out in an accordance with the manufacturer's recommendations. For this purpose, 100 g of the commercial product was placed in a beaker with a capacity of 1.5 L. Then, the product was flooded with warm water at 30 °C. After one day, 10% by volume of the inoculum containing active microorganisms was added to the cultures containing various amounts of chlorine ions.

### 2.2. Culture Condition
### 2.2.1. Dechlorination Process

The process was set up in mineral medium: $Na_2SO_4$ 4.50 g/dm$^3$, $KH_2PO_4$ 0.50 g/dm$^3$, $NH_4Cl$ 1.00 g/dm$^3$, $CaCl_2 \cdot 6\,H_2O$ 0.06 g/dm$^3$, $MgSO_4 \cdot 7\,H_2O$ 0.10 g/dm$^3$, $FeSO_4 \cdot 7H_2O$ 0.10 g/dm$^3$, and sodium citrate 0.30 g/dm$^3$. The mineral medium used in the process provided by DB was supplemented by 5.00 g/dm$^3$ of $KNO_3$. The experimental set-up was provided in glass reactors with a volume of 0.25 dm$^3$ at a temperature 23 $\pm$ 2 °C for 3 months in anaerobic conditions in darkness.

The tranformer oil with PCBs was used as the main carbon source. It was added to the culture medium at 0.5 vol%.

### 2.2.2. Chloride-Removing Process

The process was set up in two media. Medium A was an aqueous solution enriched with mineral salts: $Na_2HPO_4$ 0.20 g/dm$^3$, $KH_2PO_4$ 0.20 g/dm$^3$, $MgSO_4$ 0.02 g/dm$^3$, and $FeSO_4 \cdot 7H_2O$ 0.01 g/dm$^3$, and the pH of the solution was 7.5. Medium B was tap water with pH = 7.8. The experimental set-up was provided in glass reactors with a volume of 0.25 dm$^3$ at a temperature 23 $\pm$ 2 °C for 21 days in aerobic conditions.

As a source of chloride ions, $NH_4Cl$ was used. It was added to the culture media A and B in the following amounts: 100 mg Cl$^-$ per 1 L of medium, 200 mg Cl$^-$ per 1 L of medium, and 300 mg Cl$^-$ per 1 L of medium.

### 2.3. Transformer Oil with PCBs

The waste transformer oil used in this study originated from the Paraffin Institute of Warsaw (Poland) and consisted of 70% PCB congeners.

The analysis of the used transformer oil containing PCBs was performed by the GC/MS method on a gas chromatograph (Fisons Instruments type GC8000) with an on-column injector coupled to a mass detector (MD 800) (see Table 1). The apparatus was equipped with a 30 m long RTX-1 column, and the carrier gas was nitrogen.

### 2.4. Chloride Ion Determination
### 2.4.1. Dechlorination Process

Chemical analyses were conducted following DIN 51527 guidelines [21]. The PCB congeners were extracted in acetone/n-hexane (1:3, $v/v$), and then purification and separation were conducted by solid-phase extraction [22].

The polarographic method with a liquid mercury electrode was used for determination of chloride ions. The ECO-TRIBO Polarograf software from the Czech company Polaro-

Sensor was used [23]. The main electrode was a mercury drop electrode, and the reference electrode was a silver Ag/AgCl electrode, which was separated from the test solution by a salt bridge filled with 1M $KNO_3$, so that the sample was not contaminated with chloride ions. The electrode parameters used in the determination of chlorides are compiled in Table 2.

**Table 1.** The PCB congeners concentrations in transformer oil.

| PCB Congener | Retention Time [min] | The Congener Concentration [ppm] |
|---|---|---|
| 2,6-dichlorobifenyl | 25.104 | 368.2 |
| 4,4′-dichlorobifenyl | 26.595 | 11.2 |
| 2,5′-dichlorobifenyl | 27.135 | 36.7 |
| 2,5-dichlorobifenyl | 27.445 | 652.3 |
| 2′,3,4-trichlorobifenyl | 28.396 | 93.1 |
| 2,2′,5-trichlorobifenyl | 29.727 | 689.2 |
| 2,2′,5′-trichlorobifenyl | 29.827 | 573.4 |
| 2,4,6-trichlorobifenyl | 30.237 | 42.3 |
| 2,4,4′-trichlorobifenyl | 30.607 | 691.0 |
| 2,3′,4-trichlorobifenyl | 31.658 | 73.3 |
| 2,4′,5-trichlorobifenyl | 31.758 | 25.3 |
| 2,4,4′-trichlorobifenyl | 32.128 | 691 |
| 2,2′,3,5′-tetrachlorobifenyl | 32.588 | 630.3 |
| 2,3,4-trichlorobifenyl | 32.878 | 373.1 |
| 2,2′,4,6-tetrachlorobifenyl | 33.139 | 51.2 |
| 2,2′,5,6′-tetrachlorobifenyl | 33.479 | 19.3 |
| 2,3,4′,6-tetrachlorobifenyl | 33.879 | 163.2 |
| 2,2′,5,5′-tetrachlorobifenyl | 34.079 | 123.8 |
| 2,3,5,5′-tetrachlorobifenyl | 34.209 | 47.7 |
| 2,2′,4,4′-tetrachlorobifenyl | 34.289 | 145.3 |
| 2,2′5,6-tetrachlorobifenyl | 34.810 | 131.2 |
| 2,2′,3,4,4′,5,5′-heptachlorobifenyl | 35.859 | 134.3 |
| 2,3,4,4′-tetrachlorobifenyl | 36.910 | 41.3 |
| 2,3′,4,4′-tetrachlorobifenyl | 37.641 | 91.7 |
| 2,2′,3,4,4′,5′-heksachlorobifenyl | 37.792 | 92.7 |
| 2,3′,4,4′,5-pentachlorobifenyl | 37.891 | 62.3 |
| 2,2′,3,4,5′-pentachlorobifenyl | 38.171 | 13.2 |
| 2,2′,4,4′,5,5′-heksachlorobifenyl | 38.352 | 17.1 |
| 2,2′3,3′,5,5′-heksachlorobifenyl | 39.542 | 12.3 |

**Table 2.** Parameters of chloride determination carried out using the polarographic method.

| Preparation | Potential |
|---|---|
| Stirring: 10 s Deoxidation: 300 s Standstill: 1 s | At rest: 0 mV Initial: 300 mV Final: −200 mV Cleaning: 0 mV Electrolysis: 200 mV Oxidation: 0 mV |
| **Drop** | **Impulse** |
| Tap: 0.15 s Valve: 0.20 s | Height: 50 mV Width: 100 ms Measurement: 20% to 80% |
| **Stripping** | **Browsing** |
| Cleaning: 0 s Accumulation: 20–120 s Standstill: 30 s Oxidation: 0 s | Speed: 20 mV/s Number of cycles: 1 Standstill: 0 s |
| Current measurement accuracy: 1 nA | |

Chloride ions were determined in a medium containing the mixture: water/1M $HNO_3$/$CH_3OH$ in the ratio 10/1/10 vol%. The measurement was carried out in accordance with the parameters presented in Table 2. The test sample was taken from the aqueous phase of the culture medium. Prior to the determination of chlorides, sulfides were precipitated, which are formed in the process of anaerobic degradation in the case of sulfate reduction and may interfere with the results obtained in the analyses. The results of the chloride determination were developed using the model curve method prepared for the NaCl solution.

The effectivness of the dechlorination process was determined by the GC method. Analyses were performed on an SRI 8610 gas chromatograph (SRI INSTRUMENTS, Torrance, CA, USA) for which the detection limit was 1 bpm. The chromatograph contained an electron capture detector (ECD) and a DB-5 capillary column (30 m × 0.25 mm × 0.25 mm). The carrier gas was nitrogen. The analysis of PCB congeners was carried out with the following temperature program: the initial temperature of the furnace was 170 °C and was maintained for 5 min, and then the temperature increased linearly by 12 °C per minute to 230 °C. Further, the temperature increased by 3 °C per minute until it reached 250 °C. The temperature was held constant at 250 °C for 20 min. The Peak Simple II software version 3.86 was used for the analyses.

For identification and calibration, the Supelco PCB Congeners Mix 1 [24,25] consisted of six PCB congeners (IUPAC no. 10, 28, 52, 138, 153, 180) in concentrations of 10 mg/$dm^3$ each in iso-octane was used. The effectiveness of the dechlorination process of chosen PCB congeners was quantified by the difference between the concentration at the beginning and the end of each experiment.

### 2.4.2. Chloride-Removing Process

The measurement was performed with the spectrophotometric method. The chloride-cell test based on the APHA methods 4500-$Cl^-$ E [26]. The UV/VIS 8453 spectrophotometer Spectroquant Nova 400 (Merck KGaA, Darmstadt, Germany) with optical glass vials (Supelco 1.14946, rectangular cells 10 mm) was used to detect chloride ions at the wavelengths at λ 468 nm. The reference sample was the mineral medium A and medium B with different concentrations of chloride ions. The reference value was a blank measured before starting the cultivation.

### *2.5. Statistics*

The results obtained were analyzed statistically in STATISTICA Version 10. The presented results are the average of 5 independent biological repetitions. They were subjected to a single-factor analysis of ANOVA variance and then analysed using the Duncan multiple-range post hoc test ($p < 0.05$) to show statistically significant differences between the tested samples.

### 3. Results and Discussion

According to the data published in the literature, the process of PCB degradation in anaerobic conditions consists of the separation of chloride ions from congeners [27]. The separated chlorides accumulate in the culture medium.

In the conducted research, the amount of chlorides in the culture medium before and after the anaerobic degradation was determined. The theoretical maximum amount of chlorides was calculated on the basis of chromatographic analyses (Table 1). It was 2585.22 ppm. After the biological processes, 193.39 ppm of chloride ions were indicated in the culture medium provided by the sulphate-reducing bacteria (SRB) and 236.07 ppm in the cultures with a presence of denitryfing bacteria (DB). The amount of chlorides that were separated from PCB molecules gave adequately 7.48% for the SRB and 9.13% for the DB of decreasing chloride ions as compared to the initial value. The determination of ion quantity confirmed the course of the dechlorination process.

Table 3 shows the effectivnes of the dechlorination process of the chosen congeners in the cultures provided by SRB and DB present in the primary sediments that originated from the petrochemical wastewater sedimentation tank. The results were compared to the chosen PCB congener initial concentrations, which are presented in Table 1.

**Table 3.** The degradation of chosen PCB congeners.

| IUPAC PCB Name | SRB | | DB | |
|---|---|---|---|---|
| | Concentration after Process [ppm] | Degradation [%] | Concentration after Process [ppm] | Degradation [%] |
| 2,6-CB (10) | 198.4 | 46.12 | 209.10 | 43.21 |
| 2,4,4′-CB (28) | 447.90 | 35.18 | 413.4 | 40.17 |
| 2,2′,5,5′-CB (52) | 96.90 | 21.71 | 102.50 | 17.21 |
| 2,2′,4,4′,5,5′-CB (138) | 14.50 | 15.20 | 11.90 | 30.41 |
| 2,2′,3,4,4′,5-CB (153) | 32.30 | 65.16 | 43.40 | 53.18 |
| 2,2′,3,4,4′,5,5′-CB (180) | 39.90 | 70.29 | 32.00 | 76.17 |

In the cultures with the use of SRB and DB microorganisms, the highest degree of PCB degradation was noted for 7-CB, which exceeded 70%. Slightly weaker results were obtained for 6-CB, congener number 153. In all cultures, good degradation effects were observed for 2-CB and 3-CB (over 35%). It is a significant result, as usually the degradation of less-chlorinated congeners focuses on the ring's fissure and the process dominates in aerobic conditions [28]. The separation of chlorides from highly chlorinated PCB congeners, such as 6-CB or 7-CB, has been the best documented mechanism of PCB degradation in anaerobic conditions to date [29,30]. The determination of the amount of chlorides in the post-culture medium proves the existing hypothesis of the dechlorination process. Khalid et al. [31] briefly described the anaerobic PCB degradation pathway, which proceeds by removing chlorines in high-chlorinated molecules from meta positions and then continues to the ortho position of biphenyl rings. The PCB metabolite intermediates activate the expression of enzymes in the reductive dechloriantion process.

Even if the chloride ions separate from the biphenyl rings they still exist in the environment. The concentration of ions increase, which can cause a negative impact on ecosystems. Fortunately, there are microorganisms which are able to live in the presence of chlorides which stimulate their growth [32,33]. They provide a chloride moving process to decrease this element in polluted sites.

Some strains of halophilic bacteria are able to cope with the osmotic pressure caused by a high concentration of chloride ions in the cytoplasm, and the proteins they produce are stable and active in the presence of high concentrations of these ions [34,35].

The second process described in this paper focused on the elimination of chloride ions from polluted sites. This process is a consequence of the previous one, which can appear in polluted sites. Figure 1 presents the average concentrations of chloride ions for the control tests and the tests performed after the first, second, and third week of the experiments in two media: (a) aquatic medium with mineral salts and (b) tap water.

The effect of the chloride ion concentration changes was observed. The obtained results are presented in Table 4.

The analysis of the obtained results shows that the decrease of chloride ions in the aquatic environments did not exceed 16%. The results depended on the initial ion concentration and the most significant decrease was observed in the samples with 100 $Cl^-$ $mg/dm^3$ after 21 days of cultivation. The results proved that the use of the microorganism mixture was able to move chlorides from environment. Nevertheless, the environmental effect was not satisfied. It was also observed that the loss of chlorides in the environment depended on the time of culture.

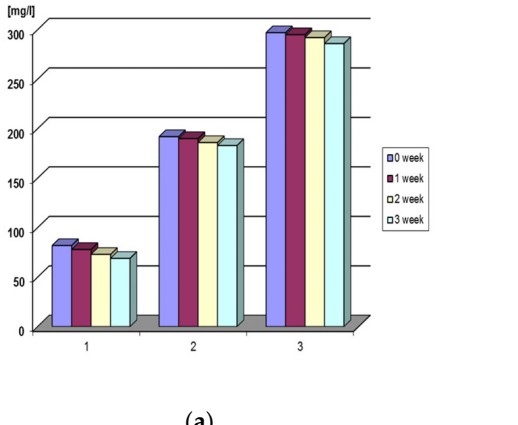
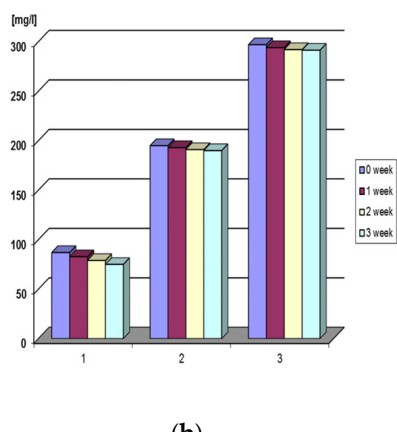

(**a**)                                                    (**b**)

**Figure 1.** The decrease in chloride ion concentration: (1) 100 mg/L; (2) 200 mg/L; (3) 300 mg/L after 21 days of cultivation by commercial mixture BIOACTIV PH.

**Table 4.** The chloride ion movement during the process of biodegradation.

| Medium | $Cl^-$ [mg/dm$^3$] | Chlorides Decrease [%] | | |
|---|---|---|---|---|
| | | **1st Week** | **2nd Week** | **3rd Week** |
| A | 100 | 4.88 | 10.98 | 15.85 |
| | 200 | 1.04 | 3.13 | 4.69 |
| | 300 | 0.67 | 1.68 | 2.17 |
| B | 100 | 4.83 | 9.20 | 13.79 |
| | 200 | 1.03 | 2.05 | 2.56 |
| | 300 | 1.01 | 1.68 | 2.36 |

There was also a difference in the loss of chloride ions depending on the medium used. The amount of chloride loss in medium B, which was tap water, was lower than in medium A, which contained mineral salts. This difference is appreciable for the samples with a chloride concentration of 100 mg/dm$^3$. The chloride degradation level in the samples where the initial chloride concentrations were 200 mg/dm$^3$ and 300 mg/dm$^3$ can be described as imperceptible. A slight change was visible for the samples from week 3.

The loss of chloride ions was more effective with the mineral medium A than in the samples containing water. A greater efficiency of chloride removal with the use of a commercial mixture in a medium with mineral salts may indicate good nutritional conditions, as well as more favourable conditions affecting the viability of microorganisms, which translates into a better and more effective degradation of compounds containing chlorine compounds in an aquatic environment.

The effect of the chloride ion concentration changes suggested that they could be calculated by a linear function. The model is described with the equations presented in Table 5. The high value of the correlation coefficients $R^2$ (Table 5) confirmed the proper use of the linear functions in the model description.

The final effect of the study resulted from the inclusion of chloride ions in the metabolic pathways of microorganisms. The chloride ions could be the electron donors, while the minerals included in the aquatic media were the electron acceptors. The redox reactions which appeared in the environment determined the microorganisms' opportunity to eliminate chlorides. It is also well known that chlorine and its ions are used as the components of disinfection products that stop the physiological functions of microorganisms. This is a reason for the low environmental effect of chlorides moving from the aquatic media.

The commercial mixture of microorganisms BIOACTIV PH is intended for the degradation of compounds such as phenols, formalin, benzene, chloroethane, and chlorinated hydrocarbons. It is mainly used in the chemical, petrochemical, and cooking industries.

However, the presented study proved its opportunity to remove chloride compounds in an aquatic environment.

**Table 5.** The equations and correlation coefficients for the model describing changes in chloride ion concentration.

| Medium | $Cl^-$ [mg/dm$^3$] | Equation | $R^2$ |
|---|---|---|---|
| A | 100 | $f_{100}(x) = -4.4x + 86.5$ | 0.9979 |
|  | 200 | $f_{200}(x) = -3.1x + 195.5$ | 0.9856 |
|  | 300 | $f_{300}(x) = -2.4x + 299.5$ | 0.9931 |
| B | 100 | $f_{100}(x) = -3.98x + 90.9$ | 0.9996 |
|  | 200 | $f_{200}(x) = -1.7x + 196.5$ | 0.9797 |
|  | 300 | $f_{300}(x) = -1.88x + 298.3$ | 0.9243 |

## 4. Conclusions

Chlorine is one of the macroelements present in living organisms, it being the main component of their physiological solution. It is a fundamental ingredient in animal and plant nutrition. It appears in organic and inorganic compounds in nature, and it is necessary for the proper functioning of ecosystems. Nevertheless, if its natural balance is destabilized by an increase in chloride concentration, a negative effect can be observed in the environment.

There are different methods which allow to restore a natural balance of chlorides in ecosystems.

In biological processes, microorganisms can remove chlorides from pollutants to decrease the toxicity of compounds (dechlorination). Other consortia of microorganisms, such as halophilic bacteria or halophilic filamentous fungi, are able to decrease chloride amounts from the environment by using the ions in their metabolic pathways.

The first investigated process concerned reductive dechlorination in anaerobic conditions. It was provided by various consortia of microorganisms originating from the petrochemical wastewater sedimentation tank (Door's tank) of a petrochemistry plant. Sulphate-reducing bacteria (SRB) and denitryfying bacteria (DB) were used in this research.

In this process, the decrease in toxic molecules was obtained in a range of 15% to 76%. The highest degree of PCB degradation was noted for 7-CB (over 70%). Good degradation effects were observed for 2-CB and 3-CB (over 35%). It is a significant result, as usually the less-chlorinated PCB congeners are degraded in aerobic condition in the ring's fissure reactions.

The second described biological process presented the results of the research with the BIOACTIVE PH commercial mixture. The effectiveness of the chloride removal from environment was estimated at 16%. It is a useful product that can eliminate toxic ions. The investigated commercial mixture can support other techniques used in soil and water remediation to decrease chlorine and its compounds in a polluted environment.

**Funding:** This research received no external funding.

**Institutional Review Board Statement:** Not applicable.

**Informed Consent Statement:** Not applicable.

**Data Availability Statement:** The data are contained within the article.

**Conflicts of Interest:** The author declares no conflict of interest.

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
