# Peer review of "The Biological Processes of Chloride Ions Removal from the Environment"

_applsci, doi:10.3390/app12178818_

Round 1
Reviewer 1 Report
Dear Authors,
some remarks to the good work so far done:
1) aim specification and structural improvements necessary so the reader understands exactly, what are the aim, tasks and how results derive to the conclusions
2) Introduction part might be improved, suggest adding references on contaminated surface waters where improvements might help by the research given. E.g., Contributions of DOC from surface and groundflow into Lake Võrtsjärv (Estonia) by Tamm et al., or Urban hydrology research fundamentals for waste management practices by Pehme et al.
3) minor improvements of Figures to have better resolution
Author Response
Dear Sir,
Thank you very much for your comments and suggestions.
Dear Authors,
some remarks to the good work so far done:
1) aim specification and structural improvements necessary so the reader understands exactly, what are the aim, tasks and how results derive to the conclusions
Thank you for the comments. I read the manuscript again and I changed the text to make it reader friendly.
2) Introduction part might be improved, suggest adding references on contaminated surface waters where improvements might help by the research given. E.g., Contributions of DOC from surface and groundflow into Lake Võrtsjärv (Estonia) by Tamm et al., or Urban hydrology research fundamentals for waste management practices by Pehme et al.
Thank you for the comments. I improved the Introduction on contaminated surface waters adding the information described in the following papers.
Tamm, T., Noges, T., Jarvet, A., Bouraoui, F. Contributions of DOC from Surface and Groundflow into Lake Vortsjarv (Estonia). Hydrobiologia 2008, 599, 1, 213-220.
Pehme, K-M., Burlakovs, J., Kriipsalu, M., Pilecka, J., Grinfelde, I., Tamm, T., Jani, Y., Hogland, W. Urban Hydrology Research Fundamental for Waste Management Practices. Rural and Environmental Engineering 2019, 1, 160-167.
Since years the agglomeration practices influenced the water irrigation changes or leachate recirculation changes in an urban and an agricultural hydrological cycle. Actually new challenges appear in urban and landfill management. Advanced technologies and modelling approaches allow to create the hydrological modelling tools to mitigate contamination of groundwater and deep aquifers by various ions like chlorides [4]. The organic and inorganic pollutants can influence directly to the changes of the elements like carbon, oxygen, nitrogen or phosphorous. Dissolved organic carbon is one of the most important element for microorganisms in their metabolic pathways. In contaminated sites the balance between the macro- and microelements can be destroyed if the concentrations of pollutants increase. Also the changes in amounts of dissolved organic carbon from surface and groundflow water may influence significantly to biogeochemical cycles [5].
3) minor improvements of Figures to have better resolution
Thank you for the suggestion. I will send separately the files with the figures to the Editor’s team to have better resolution of them.
Reviewer 2 Report
In this work, the purification of wastewater from chloride ions with the help of bacteria was studied.
After reading the manuscript, here my comments:
1. The research is about removing chloride ions with the help of bacteria, but author used sulphate reducing bacteria (SRB) and denitrifying bacteria (DB). Please explain why these bacteria are used for removing chloride ions in the introduction.
2. Please revise Abstract and Conclusion part of the paper. It is very difficult to understand the research from Abstract and Conclusion.
3. Please cite and compare your results following water treatment papers.
https://doi.org/10.3390/gels7040234
https://doi.org/10.3390/polym12102405
After revising according to the above mentioned comments, I recommend that this article be published.
Wish you success and good luck in your publishing career.
Author Response
Dear Sir,
Thank you for your suggestions which allowed me to improve the manuscript.
In this work, the purification of wastewater from chloride ions with the help of bacteria was studied.
After reading the manuscript, here my comments:
- The research is about removing chloride ions with the help of bacteria, but author used sulphate reducing bacteria (SRB) and denitrifying bacteria (DB). Please explain why these bacteria are used for removing chloride ions in the introduction.
Thank you very much for the suggestion. I wrote few more sentences in Introduction (page 2) which explain the use of the sulphate reducing bacteria and denitrifying bacteria in the described experiments as the microorganisms commonly known in anoxic degradation of chlorinated compounds like polychlorinated biphenyls.
There are many species of bacteria which provide the biodegradation processes in polluted sites. Some of them use various ions to provide the redox reactions in deeper level of the soil where the oxygen is limited. The most common microorganisms participated in dechlorination process are sulphate reducing bacteria (SRB) and denitrifying bacteria (DB) [16].
- Please revise Abstract and Conclusion part of the paper. It is very difficult to understand the research from Abstract and Conclusion.
Thank you very much for the comments. I rewrote both of the parts of the submitted manuscript.
The previous version of Abstract
The ability of chloride ions removing by sulphate reducing bacteria and denitrifying bacteria originated from the primary sediment as well as the microorganisms included in commercial mixture was tested. Two biological processes were investigated. The first one was the reductive dechlorination process in an anaerobic condition which separated chlorides from polychlorinated biphenyls included in a transformer oil. This allowed to convert hazardous molecules to less toxic compounds and influence significantly the contamination since 15 to 76%. In the second process various chlorides concentrations in two aquatic solutions were investigated. The process was carried out by microorganisms originated from biological commercial mixture in aerobic condition. The decreasing of chloride ions did not exceed 14%. Both the consortia of microorganisms and biological commercial mixture used in the studies were able to decrease the chloride ions concentration from the solution.
A new verion of Abstract
Chlorine is one of the macronutrients commonly found in nature. The natural cycle of this element can be destabilized by human activities and causes negative effects in the environment. To come back into natural balance various biological processes of water and soil remediation are investigated. The purpose of the presented research focused on two chlorine conversion processes: 1) anaerobic dechlorination of polychlorinated biphenyls (PCBs) included in transformer oil provided by consortia of microorganisms originated from wastewater sedimentation tank and 2) chlorides elimination from aquatic environment by commercial mixture of microorganisms in aerobic condition.
The dechlorination allowed to convert PCB molecules to less toxic compounds and influence significantly the contamination since 15 to 76%. In the second process the decreasing of chloride ions did not exceed 14%. Both the consortia of microorganisms and biological commercial mixture used in the studies were able to decrease the chloride ions concentration from the investigated solution.
The previous version of Conclusions
Chloride ions appear in the environment in the forms of organic or inorganic compounds. They are often a part of the compounds polluting the ecosystems, such as PCBs, or accumulate due to excessive use, such as salt spreading in winter. In the environment, the balance of this element should be maintained between the forms of its occurrence. Naturally existing microorganisms help to maintain this balance by carrying out various processes that turn chlorides into cellular responses.
One of the biological processes is the reductive dechlorination in anaerobic condition provided by various consortia of microorganisms. This process is effective and decrease the aquatic or soil environment from the contamination toxic compounds. The second experiment of presented studies found that the BIOACTIVE PH commercial mixture, which is offer as the product to some groups of the hydrocarbons utilization from the environment, can be used in chloride ions movement. It is a useful product that can eliminate toxic ions influencing the lives of fauna and flora. The investigated commercial mixture can support the other techniques used in soil and water remediation to decrease chlorine and its compounds in the polluted environment.
A new verion of Conclusion
Chlorine is one of the macroelements present in living organisms being the main component of the physiological solution. It is a fundamental ingredient in animals and plants nutrition. It appears in organic and inorganic compounds in nature and it is necessary for the proper functioning of ecosystems. Nevertheless, if its natural balance is destabilized by increasing of the chlorides concentration the negative effect can be observed in environment.
There are different methods that allow to achieve a natural balance of chlorides in ecosystems again.
In biological processes the microorganisms can remove chlorides from pollutants to decrease the toxicity of the compounds (dechlorination). The other consortia of microorganisms are able to decrease the chlorides amounts from environment using the ions in their metabolic pathways halophilic bacteria or halophilic filamentous fungi.
The first investigated process concerned the reductive dechlorination in anaerobic condition. It was provided by various consortia of microorganisms originated from the petrochemical wastewater sedimentation tank (Door’s tank) of Petrochemistry Plant. The Sulphate Reducing Bacteria (SRB) and Denitryfying Bacteria (DB) were used in the reserach.
In this process the decrease of toxic molecules was obtained in a range of 15 to 76%. The highest degree of PCB degradation was noted for 7-CB (over 70%). The good degradation effects were observed for 2-CB and 3-CB, over 35%. It is a significant information as the less chlorinated PCB congeners are degraded in aerobic condition in the rings fissure reactions.
The second described biological process presented results of the research with the BIOACTIVE PH commercial mixture. The effectiveness of the chlorides removal from environment was estimated on 16%. It is a useful product that can eliminate toxic ions. The investigated commercial mixture can support the other techniques used in soil and water remediation to decrease chlorine and its compounds in the polluted environment.
- Please cite and compare your results following water treatment papers.
Akhmetzhan, A., Myrzakhmetova, N., Amangeldi, N., Kuanyshova, Z., Akimbayeva, N., Dosmaganbetova, S., Toktarbay, Z., Longinos, S.N. A Short Review on the N,N-Dimethylacrylamide-Based Hydrogels. Gels 2021, 7, 234. https://doi.org/10.3390/gels7040234
Baimenov, A., Berillo, D., Azat, S., Nurgozhin, T., Inglezakis, V. Removal of Cd2+ from Water by Use of Super-Macroporous Cryogels and Comparison to Commercial Adsorbents. Polymers 2020, 12, 2405. doi:10.3390/polym12102405
Thank you for the suggestion. I cited the following papers in Introduction writing them as the examples of the chemical techniques used for water remediation from metal ions.
The chemical methods used in water clean processes from toxic compounds focus on the investigation of adsorption technique which offers good results for heavy metal ions removal. Many adsorbent hydrogels like N,Ndimethylacrylamide (DMAA)-based hydrogels were reported as the effective tool in such processes [18,19]. The physical methods also can be used in soil and water remediation but high cost of the apparatus decide that this method is not popular.
Round 2
Reviewer 1 Report
hi, just minor remark. Reference nr 5 should be "Pehme", now it is incorrect "Pheme". Please fix this detail
Author Response
Dear Sir,
Thank you very much for your comments.
Hi, just minor remark. Reference nr 5 should be "Pehme", now it is incorrect "Pheme". Please fix this detail.
I would like to apologize for this mistake. I corrected the reference nr 5.
Reviewer 2 Report
Dear author,
I am satisfied with the revised revision of the manuscript. Therefore, I recommend this manuscript for the publication.
Author Response
Dear author,
I am satisfied with the revised revision of the manuscript. Therefore, I recommend this manuscript for the publication.
Dear Sir,
Thank you very much for your comments and the acceptance of the submitted manuscript.